# Waveform Optimization of Compressed Sensing Radar without Signal Recovery

**Quanhui Wang [1] and Ying Sun [2],\***

[1]    School of Information Engineering, Lingnan Normal University, Zhanjiang 524000, China
[2]    HuaWei Technologies CO., LTD., Shenzhen 518000, China
**\***    Correspondence: sun-ying1984@hotmail.com

**Abstract:** Radar signal processing mainly focuses on target detection, classification, estimation, filtering, and so on. Compressed sensing radar (CSR) technology can potentially provide additional tools to simultaneously reduce computational complexity and effectively solve inference problems. CSR allows direct compressive signal processing without the need to reconstruct the signal. This study aimed to solve the problem of CSR detection without signal recovery by optimizing the transmit waveform. Therefore, a waveform optimization method was introduced to improve the output signal-to-interference-plus-noise ratio (SINR) in the case where the target signal is corrupted by colored interference and noise having known statistical characteristics. Two different target models are discussed: deterministic and random. In the case of a deterministic target, the optimum transmit waveform is derived by maximizing the SINR and a suboptimum solution is also presented. In the case of random target, an iterative waveform optimization method is proposed to maximize the output SINR. This approach ensures that SINR performance is improved in each iteration step. The performance of these methods is illustrated by computer simulation.

**Keywords:** compressed sensing radar; waveform optimization; compressive signal processing; transmit waveform

---

## 1. Introduction

Waveform design is the key aspect of multiple-input multiple-output (MIMO) radar research, since the performance of MIMO radar depends on the specific signal design. The success or failure of waveform design directly affects the performance of MIMO radar. According to the different tasks of MIMO radar, radar performance can be improved by targeted MIMO waveform design. In [1], the authors provide a unified system approach that summarizes the latest results of waveform optimization and spectral compatibility requirements. The researchers in [2] investigated the optimization of the full-polarization radar transmission waveform and the receiver response to maximize either target detection or identification performance. The approach of embedding sensitive information into radar emissions by changing the waveform during each radar pulse was investigated in [3]. As a new signal acquisition paradigm, compressed sensing/sampling (CS) [4–7] can reduce the required dataset size without reducing the resolution or quality of the compressed signal. CS technology can be used in a variety of applications including radar, sonar, and imaging. Since radar target scenes usually satisfy sparse features, such as only a few aircraft in the vast sky, researchers have come to believe that compression sensing technology can be applied to radar signal processing. In the actual radar working environment, there are usually nonwhite noise interferences, such as colored noise, clutter, and so forth. Therefore, it is of great practical significance to study the optimal detection of compressed measurement signals in interference environments [1]. Since the concept of compressed

sensing radar (CSR) was originally proposed in the groundbreaking work of [6], many works on CSR have been carried out [7–16].

There are two different schemes of CSR signal processing. In the first scheme, the signal is recovered from compressed measurements and then processed by using conventional digital signal processing (DSP) technology. This scheme offers three distinct advantages [6,8–10]: (i) eliminating the matched filter in the radar receiver, (ii) reducing the sample rate required by the receiver's analog-to-digital converter (ADC), and (iii) offering a good potential for better resolution than traditional radar. However, the recovery algorithms are always indispensable in the first scheme, which involves an iterative optimization procedure and is therefore computationally expensive for long signals. In fact, signal recovery is not important in many signal processing applications. Therefore, a second scheme has been proposed to directly solve the inference problems (extracting certain information from the measurement) [11–13]. In other words, it can effectively solve inference problems such as detection, classification, estimation, and filtering problems while greatly reducing computational complexity. This paper focuses on the CSR signal detection problem without signal recovery. Our goal is to improve detection performance in the presence of colored interference and noise with known statistical properties. According to the signal detection theory, detection probability is a nondecreasing function of the signal-to-interference-plus-noise ratio (SINR) [15]. Therefore, the optimized SINR can be used as an objective function to replace optimized detection performance [17]. In addition, the SINR depends on the transmitted waveform. Therefore, the output SINR of the CSR can be improved by optimizing its transmit waveform.

There are many CSR waveform design methods in the literature [7,14,18]. Optimal linear processing in traditional radar interference environments uses waveform optimization to maximize the output signal-to-noise ratio (SNR) [18]. Scholars have also proposed distributed compressed sensing methods for complex scenes [14,19,20]. In [7], a waveform optimization algorithm based on the simulated annealing algorithm was proposed, which can generate waveforms with small target cross correlation. Since the recovery algorithm usually includes an iterative process, when the data length is long, the recovery algorithm is also computationally intensive, and the signal processing framework cannot meet the real-time requirements. However, all of these methods are considered for the first scheme. In this scheme, in order to effectively reconstruct the target scene, the cross correlation between different target responses must be small. Therefore, the waveform design problem comes down to minimizing the cross correlation between different target responses. Obviously, these methods are not suitable for waveform design problems of CSRs without signal recovery. For these reasons, we propose in this paper a waveform optimization method to optimize the SINR of the CSR without signal recovery in the scenario where the target signal is corrupted by colored interference and noise with known statistical properties. In [21], a comprehensive theory of matched illumination waveforms for both deterministic and random targets was presented. The design of matched waveforms based on the maximization of both SNR and mutual information (MI) was considered. With this as motivation, two different target models are discussed here. In the case of deterministic targets, the optimum transmit waveform is derived by maximizing the SINR, and a suboptimum solution is also presented. In the case of random targets, an iterative waveform optimization method is proposed to maximize the output SINR. This approach ensures that SINR performance is improved in each iteration step. This method converges quickly.

The following mathematical symbols are employed in this paper: $T(z)$, the target impulse; $f(n)$, a finite duration vector signal; $T_a(s)$, transfer function; $\mathbf{\Phi}(n)$, measurement operator; $\mathbf{h}$, the receiving filter vector; $x$, low-rate sequence; $\mathbf{x}$, the compressed signal; $L$, range of samples; $M$, $N$, receive nodes and pulses; $r(t)$, analog input signal; $t(n)$, target impulse response; $v(n)$, noise process; $\mathbf{f}$, discrete transmit signal vector; $\mathbf{r}$, received signal vector; $\mathbf{\Phi}$, measurement matrix; $(\mathbf{f}, \mathbf{h})$, transceiver pair; $\mathbf{\Lambda}$, diagonal matrix; $\alpha_p$, a set of weights; $\mathbf{f}_d$, vector representation of the desired waveform; $\mathbf{U}_p$, eigenvectors corresponding to the maximum eigenvalues; $\beta$, parameter for normalizing the waveform to unit energy; $\rho(\mathbf{f}, \mathbf{h})$, SINR at the receiving filter output; $R_t(m)$, target impulse response covariance; $\mathbf{R}_v$,

colored interference and noise covariance; $z^{-1}$, unit delay operator; $\sigma^2$, Gaussian noise variance; $\gamma$, compressive ratio; and $\boldsymbol{\Gamma}$, target transfer matrix. The rest of the paper is organized as follows. Section 2 builds the signal model of the CSR with no signal recovery and formulates this problem. The optimal solution and the suboptimal solution are derived in Section 3 to optimize the SINR of the deterministic target. Section 4 introduces an iterative method for calculating the optimal waveform and receive filter for a stochastic target. Section 5 provides several numerical examples to illustrate performance improvements. The conclusion is given in Section 6.

## 2. Problem Formulation

Let us consider the following simple one-dimensional, monostatic, single-pulse CSR. Figure 1a displays the signal model of the CSR used in this paper. Note that the module of signal recovery is not involved in this signal model because this paper only focuses on directly solving the signal detection problem in the compressed measurement domain. As shown in Figure 1a: (1) The finite duration vector signal $f(n)$ is converted to analog waveforms and is modulated and transmitted. (2) The waveform is reflected back by the target with transfer function $T_a(s)$. (3) In the receiver, the reflected waveform corrupted by additive colored interference and noise is received and demodulated. Colored interference is an additive circularly symmetric complex-valued white Gaussian noise with zero-mean, which is modeled as independent of the signal [22]. To alleviate the pressure on the conventional ADC, an analog-to-information converter (AIC) is used and then the analog signal with a large bandwidth is compressed to a discrete, low-rate sequence $x$. From a mathematical point of view, AIC can be represented by the measurement operator $\boldsymbol{\Phi}(n)$, which is the collection of $K$ sampling waveforms $\{\varphi_k(n)\}_{k=1}^K$, as shown in Figure 1c (i.e., $\boldsymbol{\Phi}(n) \triangleq (\varphi_1(n), \varphi_2(n), \cdots, \varphi_k(n))^T$). (4) Then, the compressed signal $\mathbf{x}$ is processed by a vector $\mathbf{h}$ consisting of the receiving filter to further determine the existence of the target. For each pulse repetition interval, $L$ range samples are collected to cover the range interval. With $M$ receive nodes and $N$ pulses, the received data for one coherent processing interval comprises $L \times M \times N$ complex samples.

$$T(z) = \sum_{n=0}^{L} t(n)z^{-n} \tag{1}$$

where $t(n)$ denotes the target impulse response. The noise process $v(n)$ consists of additive colored interference and Gaussian white noise, which is modeled as a wide-sense stationary (WSS) random process with known variance:

$$R_v(m) \triangleq E[v(n)v(n-m)]^{\mathrm{H}}. \tag{2}$$

This assumption was also made in [22–24].

The equivalent signal model of CSR is illustrated in Figure 2. In Figure 2, the $M \times 1$ vector $\mathbf{f}$ and the $N \times 1$ vector $\mathbf{r}$ denote the discrete transmit signal vector and the received signal vector, respectively, which are defined as

$$\mathbf{f} \triangleq [f(0) \;\; f(1) \;\; \cdots \;\; f(M-1)]^T \tag{3}$$

$$\mathbf{r} \triangleq [r(0) \;\; r(1) \;\; \cdots \;\; r(N-1)]^T. \tag{4}$$

So, we can get the received signal:

$$\mathbf{r} = \mathbf{Tf} + \mathbf{v} \tag{5}$$

where

$$\mathbf{v} \triangleq [v(0) \;\; v(1) \;\; \cdots \;\; v(N-1)]^T. \tag{6}$$

**T** is the Toeplitz matrix representing the target of the form:

$$
\mathbf{T} \triangleq
\begin{bmatrix}
t(0) & 0 & \cdots & 0 \\
t(1) & t(0) & \ddots & \vdots \\
\vdots & t(1) & \ddots & 0 \\
t(L) & \vdots & \ddots & t(0) \\
\vdots & t(L) & \ddots & t(1) \\
\vdots & \ddots & \ddots & \vdots \\
0 & \cdots & 0 & t(L)
\end{bmatrix}.
\tag{7}
$$

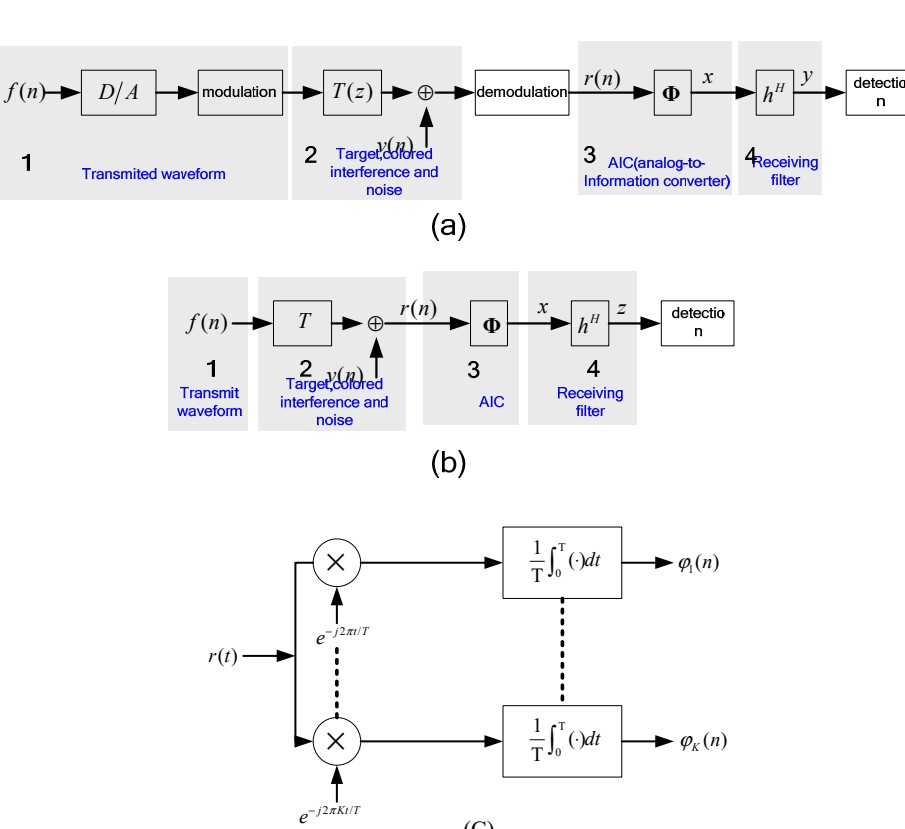

**Figure 1.** Illustration of (**a**) the signal model of the compressed sensing radar (CSR) and (**b**) the discrete baseband equivalent model. (**b**) illustrates the discrete baseband equivalent model, where the target transfer function $T(z)$ can be assumed to be a finite impulse response (FIR) filter of the form. (**c**) illustrates the $K$ sampling waveforms $\{\varphi_k(n)\}_{k=1}^{K}$, where $r(t)$ is the analog input signal.

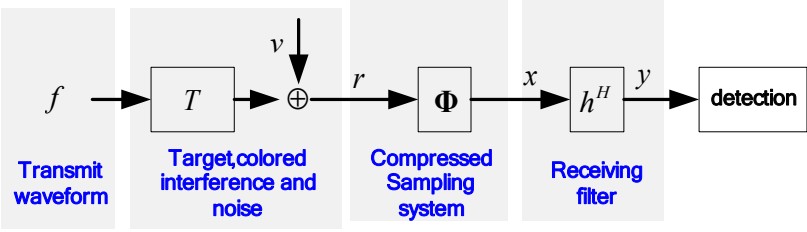

**Figure 2.** The equivalent signal model.

Thereafter, the **r** is processed by the compressed sampling module to reduce the data size. As illustrated in Figure 2, the $K \times N(K \ll N)$ measurement matrix $\mathbf{\Phi}$ is the discrete counterpart of the analog $\mathbf{\Phi}(n)$, which can be represented by a random matrix. The entries $\phi_{ij}$ of the random matrix $\mathbf{\Phi}$ are independent and identically distributed (i.i.d.) random variables. In order to avoid information loss, classical sampling theorem dictates that $K$ should be as large as $N$. However, CS technology allows for $K$ to be much less than $N$ as long as the signal is sparse. For the S-sparse signal, the measurement matrix $\mathbf{\Phi}$ needs to satisfy the restricted isometry property (RIP). The S-restricted isometry constant $\delta$ of the matrix $\mathbf{\Phi}$ is the smallest number, such that

$$(1 - \delta)\| \mathbf{c} \|_2^2 \leq \| \mathbf{\Phi c} \|_2^2 \leq (1 + \delta)\| \mathbf{c} \|_2^2 \tag{8}$$

holds for all S-sparse vectors **c**. Using the measurement matrix $\mathbf{\Phi}$, the $K \times 1$ compressed sampling vector **x** can be equivalently written as

$$\mathbf{x} = \mathbf{\Phi r} = \mathbf{\Phi T f} + \mathbf{\Phi} v. \tag{9}$$

Then, the compressed sampling vector **x** is processed by a $K \times 1$ receiving filter vector **h** to further determine the existence of the target. The receive filter output can be expressed as

$$y = \mathbf{h}^H \mathbf{x} = \underbrace{\mathbf{h}^H \mathbf{\Phi T f}}_{\text{signal}} + \underbrace{\mathbf{h}^H \mathbf{\Phi} v}_{\substack{\text{interfernce} \\ \text{and noise}}} \tag{10}$$

where

$$\mathbf{h} \triangleq [h(0)\ h(1) \cdots h(K-1)]^T. \tag{11}$$

Using a priori information of the target impulse response and the second-order statistical properties of colored interference and noise, our goal is to jointly optimize the transceiver pair (**f**, **h**) of CSR to maximize the detection performance. The detection probability is a nondecreasing function of the SINR according to the signal detection theory [13]. Our goal can be substituted to maximize the SINR by designing the transceiver pair (**f**, **h**). Thus, the SINR at the filter output can be expressed as

$$\rho(\mathbf{f}, \mathbf{h}) = \frac{\left|\mathbf{h}^H \mathbf{\Phi T f}\right|^2}{E\left[\left|\mathbf{h}^H \mathbf{\Phi} v\right|^2\right]}. \tag{12}$$

Our task is how to maximize the SINR subject to the power constraint, that is,

$$\max_{\mathbf{f}, \mathbf{h}} \rho(\mathbf{f}, \mathbf{h})\ subject\ to\ \| \mathbf{f} \|^2 \leq 1. \tag{13}$$

Comparing the transceiver pair (**f**, **h**) optimization problem in the CSR to its counterpart in conventional radar, the only difference is that the CSR involves $K \times N(K \ll N)$ compressed sampling represented by the matrix $\mathbf{\Phi}$. It is worth observing that if the matrix $\mathbf{\Phi}$ is an identity matrix, the radar system illustrated in Figure 2 becomes conventional radar. For traditional radars, many classical methods have been proposed [22–24] to solve the joint optimization problem. However, these methods do not guarantee the optimum SINR in the CSR. To better interpret this point, we assume that (**f₀**, **h₀**) is the optimal transceiver pair in conventional radar. If we use **f₀** as the transmitting waveform to pass through the CSR system illustrated in Figure 2, then the $N \times 1$ column filter vector **h₀** is employed to achieve the maximum SINR. By comparing the CSR to conventional radar, the following equation holds

$$\mathbf{h}^H \mathbf{\Phi} = \mathbf{h}_0^H \tag{14}$$

where $\boldsymbol{\Phi}$ is the full row rank. For any given random matrix $\boldsymbol{\Phi}$, according to Equation (14), we have

$$\mathbf{h}^H = \mathbf{h}_0^H \boldsymbol{\Phi}^H (\boldsymbol{\Phi}\boldsymbol{\Phi}^H)^{-1}. \tag{15}$$

Because $K < N$, it is easy to draw the following conclusion:

$$\mathbf{h}^H \boldsymbol{\Phi} = \mathbf{h}_0^H \boldsymbol{\Phi}^H (\boldsymbol{\Phi}\boldsymbol{\Phi}^H)^{-1} \boldsymbol{\Phi} \neq \mathbf{h}_0^H. \tag{16}$$

If and only if $K = N$, the corresponding SINR is optimal. However, in the CSR case, we have $K < N$. Therefore, the corresponding SINR is no longer optimal.

## 3. Optimizing Methods for Deterministic Target Impulse Response

In this section, the optimum waveform $\mathbf{f}$ is derived by maximizing the SINR over all possible choices of $\mathbf{f}$. After that, a suboptimum waveform is presented to provide suboptimal SINR. To solve (13), we can first solve $\mathbf{h}$ in terms of $\mathbf{f}$. In this case, the optimization problem becomes

$$\max_{\mathbf{h}} \frac{\left|\mathbf{h}^H \boldsymbol{\Phi}\mathbf{T}\mathbf{f}\right|^2}{\mathbf{h}^H \boldsymbol{\Phi}\mathbf{R}_v \boldsymbol{\Phi}^H \mathbf{h}} \tag{17}$$

where $\mathbf{R}_v = E\left[\mathbf{v}\mathbf{v}^H\right]$, which can be obtained by using the prior second-order statistic properties defined in Equation (2). The above problem can be recast as

$$\min_{\mathbf{h}} \mathbf{h}^H \boldsymbol{\Phi}\mathbf{R}_v \boldsymbol{\Phi}^H \mathbf{h} \, subject \, to \, \mathbf{h}^H \boldsymbol{\Phi}\mathbf{T}\mathbf{f} = 1. \tag{18}$$

This is the minimum variance distortionless response (MVDR) problem [25]. The solution to this problem is

$$\mathbf{h} = \alpha (\boldsymbol{\Phi}\mathbf{R}_v \boldsymbol{\Phi}^H)^{-1} \boldsymbol{\Phi}\mathbf{T}\mathbf{f} \tag{19}$$

where $\alpha$ is a scalar satisfying the equality constraint. Substituting the above $\mathbf{h}$ back into the objective function in Equation (12), the optimization problem becomes

$$\max_{\mathbf{f}} \mathbf{f}^H \mathbf{T}^H \boldsymbol{\Phi}^H (\boldsymbol{\Phi}\mathbf{R}_v \boldsymbol{\Phi}^H)^{-1} \boldsymbol{\Phi}\mathbf{T}\mathbf{f} \, subject \, to \, \| \mathbf{f} \|^2 \leq 1. \tag{20}$$

Now, this optimization problem only has one parameter $\mathbf{f}$. To simplify the notation, we denote $\hat{\mathbf{R}} = \mathbf{T}^H \boldsymbol{\Phi}^H (\boldsymbol{\Phi}\mathbf{R}_v \boldsymbol{\Phi}^H)^{-1} \boldsymbol{\Phi}\mathbf{T}$. Then, the objective function reduces to $\mathbf{f}^H \hat{\mathbf{R}}\mathbf{f}$. Furthermore, we let $\hat{\mathbf{R}} = \mathbf{U}\boldsymbol{\Lambda}\mathbf{U}^H$, where $\mathbf{U} = [\mathbf{u}_1, \mathbf{u}_2, \cdots, \mathbf{u}_M]$ are the eigenvectors of $\hat{\mathbf{R}}$ and the diagonal elements of the diagonal matrix $\boldsymbol{\Lambda}$, $\lambda_1 \geq \lambda_2 \geq, \cdots, \geq \lambda_M > 0$ are the corresponding eigenvalues. Then, the optimal transmit waveform is the eigenvector corresponding to the largest eigenvalue of $\hat{\mathbf{R}}$.

In fact, for waveform design problems, some constraints should be considered, such as constant modulus, range resolution, and range sidelobe level. A conventional waveform such as the linear frequency modulation (LFM) waveform is a good candidate for achieving the above characteristics. To obtain the "closest" waveform in the least-squares (LS) mean to the conventional waveform, we take full advantage of the eigenvectors $[\mathbf{u}_1, \mathbf{u}_2, \cdots, \mathbf{u}_P]$ corresponding to the maximum $P$ ($P < K$) eigenvalues and then find a set of weights $\alpha_p$, such that

$$\sum_{p=1}^{P} \alpha_p \mathbf{u}_p = \mathbf{U}_p \boldsymbol{\alpha} = \mathbf{f}_d \tag{21}$$

where $\mathbf{U}_p = [\mathbf{u}_1, \mathbf{u}_2, \cdots, \mathbf{u}_P]$; $\boldsymbol{\alpha} = [\alpha_1, \alpha_2, \cdots \alpha_P]^T$; and $\mathbf{f}_d$ is a vector representation of the desired waveform. According to Equation (21), we can obtain $\alpha = \mathbf{U}_P^H \mathbf{f}_d$. Finally, the suboptimal waveform is given by

$$\mathbf{f}_{subopt} = \beta \mathbf{U}_P \alpha = \beta \mathbf{U}_P \mathbf{U}_P^H \mathbf{f}_d \tag{22}$$

where $\beta$ is the parameter for normalizing the waveform to unit energy.

## 4. Iterative Method with Random Target Impulse Response

The methods introduced in Section 3 require that the information of the target impulse $T(z)$ is determined and known. In this section, we are concerned with the case where only partial information from the target impulse response is available. The target impulse response is modeled as a WSS random process. We assume that only the covariance of the process is known, which may be unrealistic in an actual radar detection environment. In fact, the corresponding covariance is usually estimated by an auxiliary sample [26–30]. An iterative method for maximizing the SINR is derived.

We assume that $t(n)$ is a WSS process. To simplify calculations, we suppose that its covariance is known and is defined as

$$R_t(m) \triangleq E\left[t(n)t(n-m)^H\right]. \tag{23}$$

In this case, the SINR at the receiving filter output is defined as

$$\rho(\mathbf{f}, \mathbf{h}) \triangleq \frac{E\left[\left\|\mathbf{h}^H \boldsymbol{\Phi} \mathbf{T} \mathbf{f}\right\|^2\right]}{E\left[\left\|\mathbf{h}^H \boldsymbol{\Phi} \mathbf{v}\right\|^2\right]}. \tag{24}$$

We first optimize the receiving filter $\mathbf{h}$ for the situation where the transmit waveform $\mathbf{f}$ is fixed. Then, we optimize $\mathbf{f}$ under the situation where $\mathbf{h}$ is fixed. This iterative technique is used to guarantee the SINR convergence. To solve $\mathbf{h}$ in terms of $\mathbf{f}$, the optimization problem can be written as

$$\max_{\mathbf{h}} \frac{\mathbf{h}^H \boldsymbol{\Phi} \mathbf{R}_{t,f} \boldsymbol{\Phi}^H \mathbf{h}}{\mathbf{h}^H \boldsymbol{\Phi} \mathbf{R}_v \boldsymbol{\Phi}^H \mathbf{h}} \tag{25}$$

where $\mathbf{R}_{t,f} \triangleq E\left[\mathbf{T} \mathbf{f} \mathbf{f}^H \mathbf{T}^H\right]$. Define $\mathbf{L}_{v,f}^{-1} \boldsymbol{\Phi} \mathbf{R}_{t,f} \boldsymbol{\Phi}^H \mathbf{L}_{v,f}^{-H}$ and $\mathbf{b} \triangleq \mathbf{L}_{v,f}^H \mathbf{h}$ by changing variables and the optimization problem can be recast as

$$\max_{\mathbf{h}} \frac{\mathbf{b}^H \mathbf{L}_{v,f}^{-1} \boldsymbol{\Phi} \mathbf{R}_{t,f} \boldsymbol{\Phi}^H \mathbf{L}_{v,f}^{-H} \mathbf{b}}{\mathbf{b}^H \mathbf{b}}. \tag{26}$$

This is the well-known Rayleigh quotient, and the solution to the problem is the principal component of the matrix $\mathbf{L}_{v,f}^{-1} \boldsymbol{\Phi} \mathbf{R}_{t,f} \boldsymbol{\Phi}^H \mathbf{L}_{v,f}^{-H}$. Thus, solution $\mathbf{h}$ can be expressed as

$$\mathbf{h} = \mathbf{L}_{v,f}^{-H} p\left(\mathbf{L}_{v,f}^{-1} \boldsymbol{\Phi} \mathbf{R}_{t,f} \boldsymbol{\Phi}^H \mathbf{L}_{v,f}^{-H}\right) \tag{27}$$

where $p(\mathbf{L}_{v,f}^{-1} \boldsymbol{\Phi} \mathbf{R}_{t,f} \boldsymbol{\Phi}^H \mathbf{L}_{v,f}^{-H})$ denotes the principal component of matrix $\mathbf{L}_{v,f}^{-1} \boldsymbol{\Phi} \mathbf{R}_{t,f} \boldsymbol{\Phi}^H \mathbf{L}_{v,f}^{-H}$.

To solve $\mathbf{f}$ in terms of $\mathbf{h}$, the optimization problem becomes

$$\max_{\mathbf{f}} = \frac{\mathbf{f}^H \mathbf{R}_{t,h} \mathbf{f}}{\mathbf{h}^H \boldsymbol{\Phi} \mathbf{R}_v \boldsymbol{\Phi}^H \mathbf{h}} \ subject \ to \ \| \mathbf{f} \|^2 \leq 1 \tag{28}$$

where $\mathbf{R}_{t,h} \triangleq E\left[\mathbf{T}^H \boldsymbol{\Phi}^H \mathbf{h} \mathbf{h}^H \boldsymbol{\Phi} \mathbf{T}\right]$. Please note that the denominator is temporarily scalar; therefore, we can get

$$\mathbf{f} = p\left(\mathbf{R}_{t,h}\right). \tag{29}$$

We summarize the iterative method for jointly optimizing the transmit waveform and the receiving filter in the case of the random target impulse response as follows. Given the target impulse response covariance $R_t(m)$, the colored interference and noise covariance is $\mathbf{R}_v$; for any given random matrix $\mathbf{\Phi}$, calculate the Cholesky decomposition, $\mathbf{\Phi R}_v\mathbf{\Phi}^H = \mathbf{L}_{v,f}\mathbf{L}_{v,f}^H$; initialize the transmitted waveform $\mathbf{f}$ to the conventional LFM waveform, and the transceiver pair ($\mathbf{f}$, $\mathbf{h}$) can be optimized by repeating the following steps:

(1) Compute $\mathbf{R}_{t,f} \triangleq E\left[\mathbf{Tff}^H\mathbf{T}^H\right]$.

(2) Use the resulting $\mathbf{L}_{v,f}$ to update $\mathbf{h}$ via (27).

(3) Calculate $\mathbf{R}_{t,h} \triangleq E\left[\mathbf{T}^H\mathbf{\Phi}^H\mathbf{hh}^H\mathbf{\Phi T}\right]$.

(4) Use the principal component of $\mathbf{R}_{t,h}$ to update $\mathbf{f}$ via (29).

(5) $\mathbf{f} = \mathbf{f}/\|\mathbf{f}\|$.

(6) The SINR subject to the power constraint via (13), then repeat until convergence. When SINR basically does not change much, ($\mathbf{f}$, $\mathbf{h}$) can be considered to converge. We can also set the number of iterations according to the accuracy requirements.

## 5. Numerical Results

We considered the CSR without signal recovery, as illustrated in Figure 2. The signal bandwidth was 100 MHz. Assume that the received radar signal is corrupted by a colored interference and an additive circularly symmetric complex-valued white Gaussian noise with zero-mean and variance $\sigma^2 = 1$. We borrowed the colored interference simulation model in [22], namely, the colored interference is an autoregressive (AR) random process obtained by passing a circularly symmetric complex-valued white Gaussian noise with zero-mean and variance $\sigma^2$ through the following filter.

$$H(z) = \frac{1}{\left(1 - 1.5z^{-1} + 0.7z^{-2}\right)^4} \tag{30}$$

where $z^{-1}$ denotes the unit delay operator. The random $K \times N$ matrix $\mathbf{\Phi}$ is populated with i.i.d. zero-mean Gaussian entries (of any fixed variance). Then, the orientation of the row space of $\mathbf{\Phi}$ has a random uniform distribution.

Next, we performed computer simulations considering two aspects. In this study, the parameters of the simulation platform were a CPU Intel i3 2.0 GHz with 4 GB of memory and an operating environment of MATLAB R2009b. First, a simulation example is provided for the case of a deterministic target impulse response. The SINR performance of the proposed method was compared to a conventional LFM waveform. The LFM waveform is designed to obtain a sharp ambiguity function, which is a good candidate for distinguishing two-point targets. However, in complex environments involving color interference, LFM waveforms may not have good SINR performance. Second, we provide a simulation example for the case of a target with a random impulse response.

$$S(t) = rect\left(\frac{t}{\tau}\right)e^{j\pi\mu t^2} \tag{31}$$

where $\tau$ is the plus width and $\mu$ is the slope.

### 5.1. Deterministic Target Impulse Response

First, we considered the simplest example in the case where the target model is a point target. Then, the target transfer matrix $\mathbf{\Gamma}$ became an $M \times M$ identification matrix. Monte Carlo simulation was applied to illustrate the relationship between the output SINR and the compressed ratio $\gamma$ ($\gamma \triangleq K/N$). The simulation was performed by averaging among 1000 different colored interferences and white noise implementations. We delivered four different waveforms through the CSR system, including the design results for the optimal and suboptimal waveforms presented here, as well as the traditional LFM

waveforms. Figure 3 shows the comparison of the SINR under various $\gamma$. One can see that the optimal waveform had the best SINR performances among all of the methods under all $\gamma$. The suboptimal waveform had better SINR performances than the LFM waveform in the low compressed ratio region, but the advantage gradually subsided when $\gamma$ increased. Its one execution time was almost 22 s.

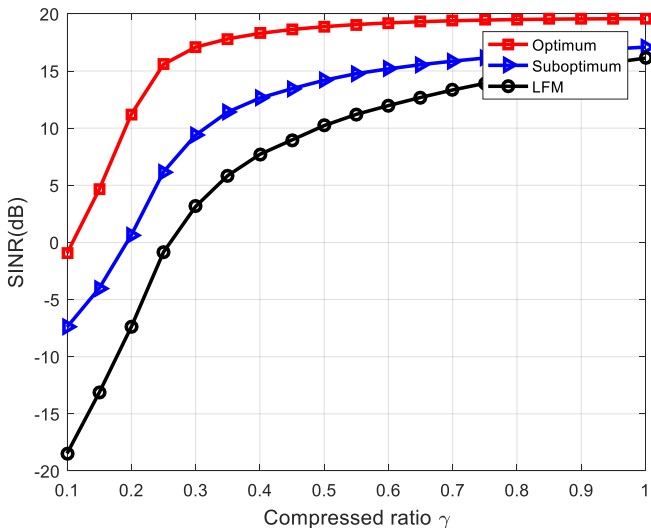

**Figure 3.** Comparison of the signal-to-interference-plus-noise ratio (SINR) versus the compressive ratio $\gamma$.

SINR improvement is equal to SINR-optimal or SINR-suboptimal minus SINR-LFM. The comparison of the SINR improvement under various $\sigma^2/\sigma_0^2$ is illustrated in Figure 4. As we can see from Figure 4, the SINR improvements provided by optimal and suboptimal waveforms were more significant relative to the LFM waveform. When $\gamma = 0.2$ and $\sigma^2/\sigma_0^2 = -20$ dB, the optimal and suboptimal waveforms provided approximately an 18 and 8 dB SINR improvement relative to the LFM waveform, respectively. Note that the SINR improvement decreased as $\gamma$ increases. Its one execution time was almost 21 s.

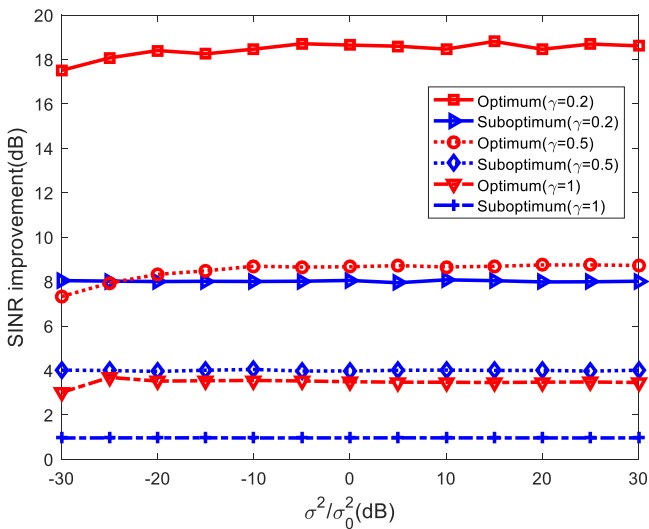

**Figure 4.** Comparison of the SINR versus $\sigma^2/\sigma_0^2$ under various compressed ratios $\gamma$.

Next, we considered two specific optimized transmit waveforms in the case where the compressed ratio $\gamma = 1$ and $\gamma \neq 1$, respectively. Figure 5a,b show a comparison of the waveform spectra of optimal and suboptimal waveforms when $\sigma^2/\sigma_0^2 = -20$ dB and $\gamma = 1$. As we expected, both the spectra

of the optimal and suboptimal waveforms had notches corresponding to the peak locations of the interference spectrum, which means that the transmit waveform energies were still concentrated in the interference-free regions of the system bandwidth. The corresponding waveform series are shown in Figure 6a,b. It is important to note that when $\gamma = 1$, the optimal waveform failed to work properly, while the suboptimal waveform seemed to give a reasonable performance, such as the range resolution and side lobe level.

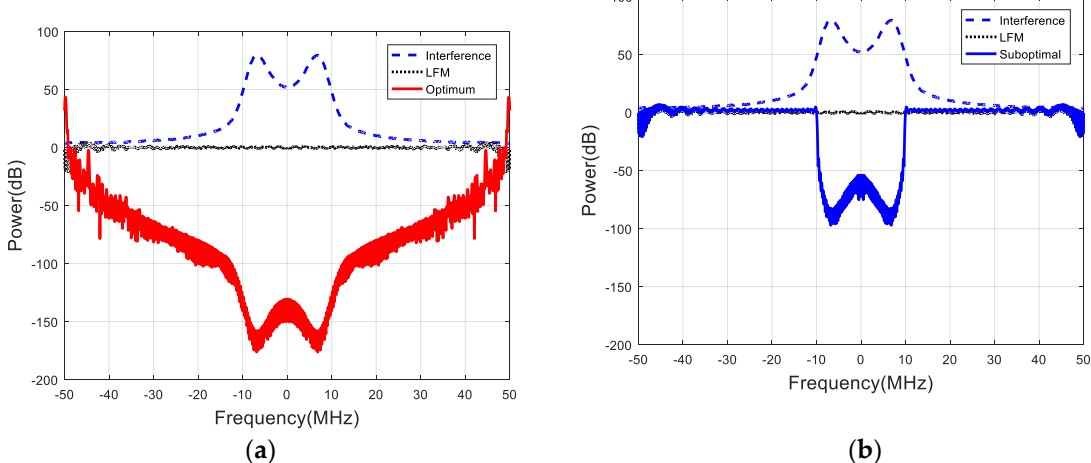

**Figure 5.** The spectra of (**a**) optimal and (**b**) suboptimal waveforms when $\gamma = 1$.

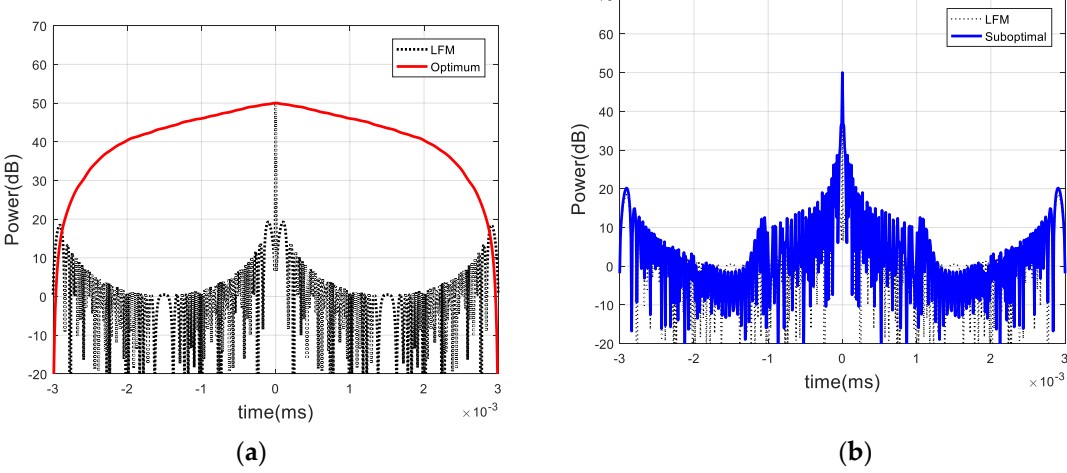

**Figure 6.** Corresponding autocorrelation functions of (**a**) optimal and (**b**) suboptimal waveforms when $\gamma = 1$.

Next, we investigated the case where the compressed ratio $\gamma \neq 1$. For one example, we assumed that $\gamma = 0.2$. Figure 7 illustrates the optimized waveform spectra's comparison with the LFM waveform spectra when $\sigma^2/\sigma_0^2 = -20$ dB. As can be seen from Figure 7a,b, both the spectra of the optimal and suboptimal waveforms also had notches corresponding to the zenith locations of the interference spectrum. Figure 8a,b show their corresponding waveform sequences. It is worth mentioning that the optimum waveform provided a reasonable side lobe level and range resolution as well, which was different from the $\gamma = 1$ case. Figure 8 shows that both optimized waveforms had reasonable distance resolutions and side lobe levels.

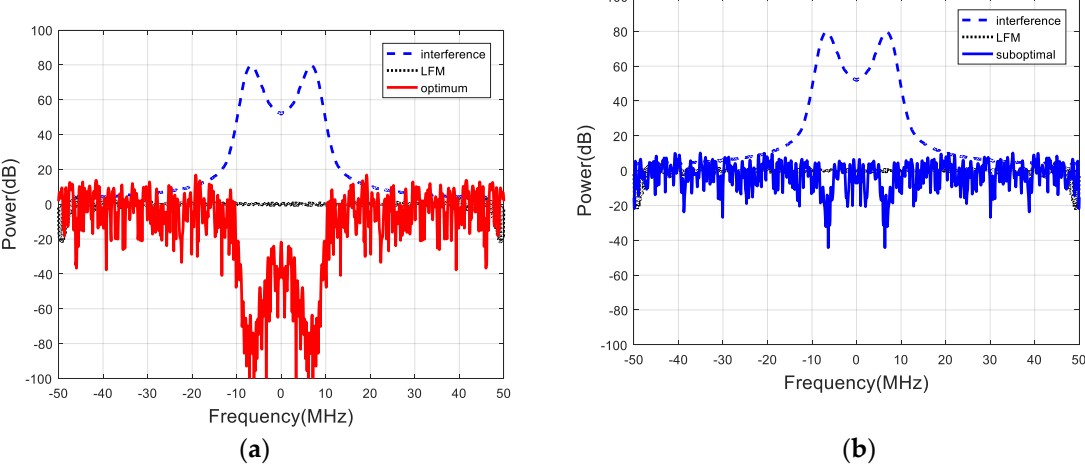

**Figure 7.** The spectra of (**a**) optimal and (**b**) suboptimal waveforms when $\gamma = 0.2$.

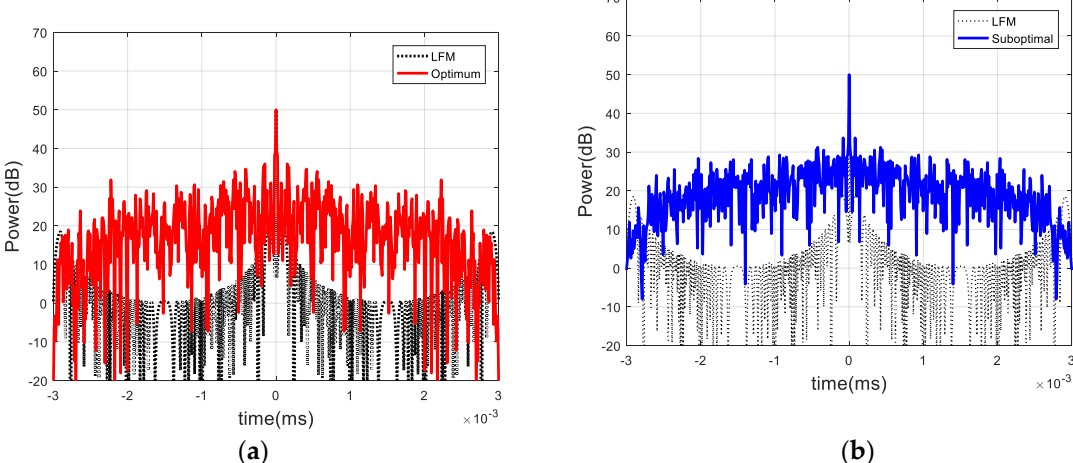

**Figure 8.** Corresponding autocorrelation functions of (**a**) optimal and (**b**) suboptimal waveforms when $\gamma = 0.2$.

Next, we considered an example of the extended target case. The target impulse response is given by

$$t(n) = \begin{cases} 1, n = 1, 7, 20 \\ 0, otherwise \end{cases}. \tag{32}$$

The simulation was performed by averaging among 1000 different colored interferences and white noise realizations. The comparison of the SINR curves is shown in Figure 9 under various $\gamma$. It can be seen that both the optimal and suboptimal waveforms had significantly better SINR performance than the LFM waveform, which was consistent with the point target case. Its one execution time was almost 9 s. Figure 10 shows a comparison of the SINR under various $\sigma^2/\sigma_0^2$. It implies that the SINR improvements provided by the optimal and suboptimal waveforms decreased as $\gamma$ increased. Its one execution time was almost 12 s.

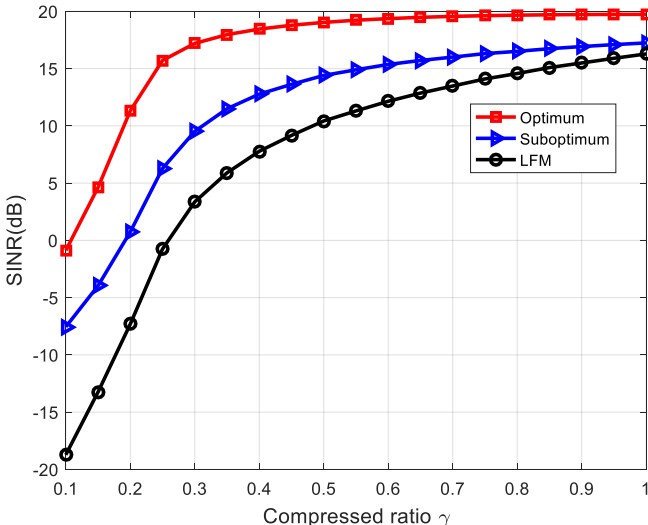

**Figure 9.** Comparison of the SINR versus the compressive ratio $\gamma$ for extended target.

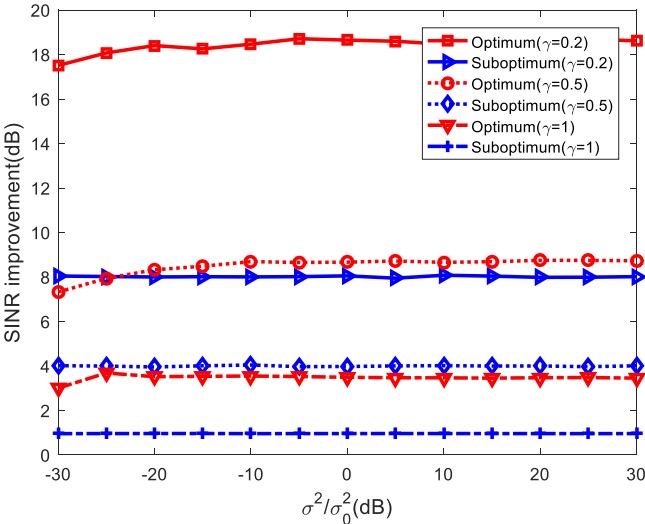

**Figure 10.** Comparison of the SINR versus $\sigma^2/\sigma_0^2$ under various compression ratios for extended target.

*5.2. Random Target Impulse Response*

This subsection describes how the coefficients of the target impulse $t(n)$ were modeled as a WSS random process with covariance $R_t(n)$. The covariance $R_t(n)$ was generated by using $R_t(n) = U_t(n) * U_t(n)^H$, where $U_t(n)$ is a sequence with a length of 20 and the coefficients $\{U_t(n)\}$ are i.i.d. circular complex Gaussian random variables. The simulation was performed by averaging among 1000 different target colored interferences and white noise realizations. The initial waveform used in the iterative method was the conventional LFM waveform. Figure 11 shows the SINR performance as a function of the number of iterations when $\gamma = 0.5$ and $\sigma^2/\sigma_0^2 = 5$ dB. One point to note here is that the LFM waveform was fixed because its SINR was not a function of the number of iterations. From Figure 11, we can observe that the iterative method converged very quickly. In this simulation example, it converged in about six iterations. As we expected, the iterative method was also superior to the LFM waveform for the random target impulse response case. Its one execution time was almost 21 s.

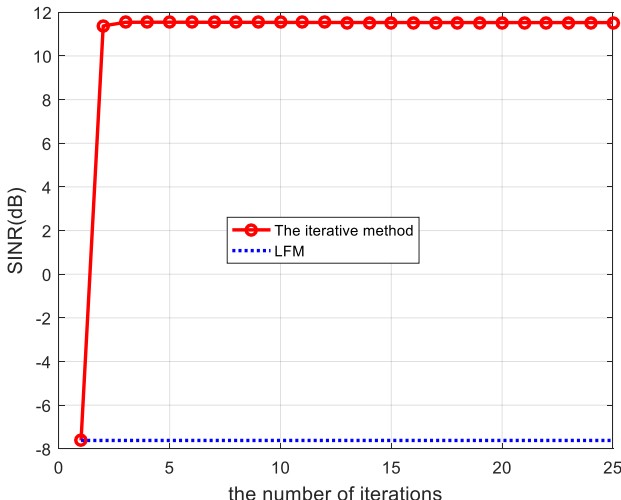

**Figure 11.** Comparison of the SINR versus the number of iterations for random target impulse response $\gamma = 0.5$.

Assume that $\sigma^2/\sigma_0^2 = -20$ dB. The SINR curve comparison between that obtained from the iterative method and the LFM waveform is illustrated in Figure 12. Its one execution time was almost 66 s. Figure 13 shows the comparison of the SINR under various $\sigma^2/\sigma_0^2$. It can be seen that the iterative method had a significantly better SINR performance than the LFM waveform. Unlike the deterministic target case, the SINR improvement increased as the compressed ratio $\gamma$ increased.

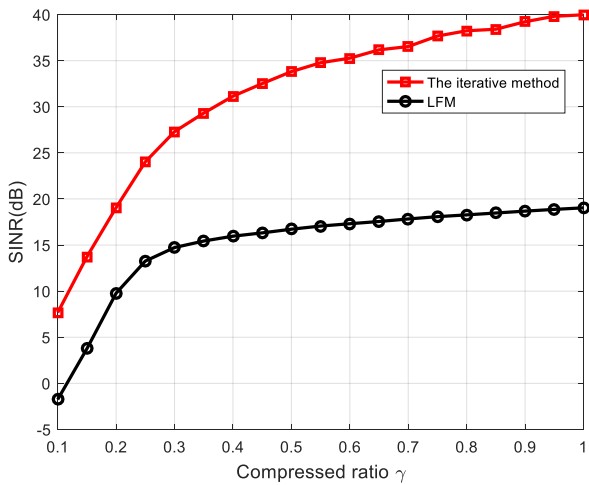

**Figure 12.** Comparison of the SINR versus compressive ratio for random target impulse response.

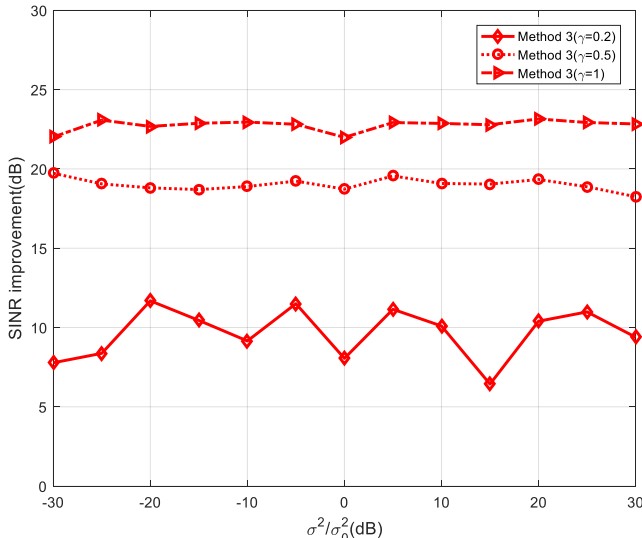

**Figure 13.** Comparison of the SINR versus $\sigma^2/\sigma_0^2$ under various compression ratios $\gamma$ for random target impulse response.

The computational complexity of the optimization approaches in both the deterministic and random cases can be expressed by the one execution time. As can be seen, the deterministic case took more time than the random case.

## 6. Conclusions

Modern radar systems often operate with large bandwidths. The resulting high sample rates according to the Nyquist–Shannon theorem cause a large computational burden and large power consumption for radar systems. CS technology provides a solution to radar signal processing problems with significantly reduced data size. The benefit of reducing the amount of data is to save memory and power consumption as well as reduce acquisition time. In this paper, we introduced a CSR without signal recovery. Our mission was to improve the performance of its target detection. The radar's waveform optimization could significantly improve the SINR required to improve target detection. This study examined the target with deterministic impulse response and the target with random impulse response. In the first case, the optimal transmit waveform was derived by maximizing the SINR and a suboptimal solution was also presented. In the second case, an iterative method of jointly optimizing the transmit waveform and the receive filter was proposed to maximize the SINR. For different target models, Monte Carlo simulations were performed to verify the effectiveness of the proposed waveform optimization method. Finally, the numerical results showed that the proposed method had a better SINR than the LFM waveform. The sparse signal compression detection results in the clutter were mainly ideal clutter models. However, in actual radar systems, training data are often used to estimate the covariance of clutter. The next step is to study the detection performance of the sparse signal compression detector to estimate the clutter covariance. AIC is the core issue of CSR, and radar performance can be further improved by optimizing AIC. The next step is to study AIC design and its hardware implementation.

**Author Contributions:** Conceptualization, investigation, supervision and writing, Q.W.; review and editing, Y.S.

**Funding:** This work was support by the National Natural Science Foundation of China: 61703280, China Postdoctoral Science Foundation: 2018M643184.

**Conflicts of Interest:** The authors declared that they have no conflict of interest to this work. We declare that we do not have any commercial or associative interest that represents a conflict of interest in connection with the work submitted.

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
