# Peer review of "Waveform Optimization of Compressed Sensing Radar without Signal Recovery"

_information, doi:10.3390/info10090271_

Round 1

Reviewer 1 Report

In the present paper the authors propose two different algorithms for radar waveform design. Precisely, they solve an optimization problem whose objective function is the receiving SINR output, moreover they exploit some sparse characteristics of the target received signal to properly select it.

In the reviewer opinion the idea is interesting, however the Reviewer has some concerns that must be solved before accepting it.

First of all, the introduction is not well written. It is not clear at all the context in which the proposed algorithm works; in fact, many references are missed as well as no related works are properly discussed. For instance, paper [1] below could be cited since it provides a unifying and systematic approach summarizing recent results on waveform optimization with also spectral compatibility requirements. Also, it is not easy to understand the novelty of the paper by reading the Introduction. This aspect should be better emphasized by the authors.

The utilized notation is always confused. It is difficult to distinguish among vectors and matrices. Sometimes vectors are bold lower case, other times they are lower case with no bold.

Which is the difference between the two equations in (8)?

Line 165: We assume that only the covariance of the process is known. Please add a footnote recalling that the covariance matrix can be estimated citing also some references that deal with this problem, e.g., [2-5].

I do not understand why the SINR improvements degrade with increasing K. Please give a motivation to this result.

I would like to see the comparison of the two proposed methods. Why in the first analyses you consider only the first algorithm, and only then the other? Moreover, since you are in the under-sampled case, i.e. K<N, what happens if you utilized the sample covariance matrix in place of the true one?

[1] A. Aubry, V. Carotenuto, A. De Maio, A. Farina, and L. Pallotta, “Optimization Theory-Based Radar Waveform Design for Spectrally Dense Environments”, IEEE Aerospace and Electronic Systems Magazine, Vol. 31, No. 12, pp. 14-25, 2016.

[2] I. S. Reed, J. D.Mallett, and L. E. Brennan, “Rapid Convergence Rate in Adaptive Arrays”, IEEE Trans. on Aerospace and Electronic Systems, Vol. 10, No. 4, pp. 853-863, November 1974.

[3] M. Steiner and K. Gerlach, “Fast Converging Adaptive Processors for a Structured Covariance Matrix”, IEEE Trans. on Aerospace and Electronic Systems, Vol. 36, No. 4, pp. 1115-1126, October 2000.

[4] A. Aubry, A. De Maio, and L. Pallotta, “A Geometric Approach to Covariance Matrix Estimation and its Applications to Radar Problems", IEEE Trans. on Signal Processing, Vol. 66, No.4, pp. 907-922, February 2018.

[5] A. De Maio, L. Pallotta, J. Li, and P. Stoica, “Loading Factor Estimation under Affine Constraints on the Covariance Eigenvalues with Application to Radar Target Detection”, IEEE Trans. on Aerospace and Electronic Systems, Vol. 55, No. 3. pp. 1269-1283, June 2019.

Author Response

Response to Reviewer 1 Comments

Point 1:

First of all, the introduction is not well written. It is not clear at all the context in which the proposed algorithm works; in fact, many references are missed as well as no related works are properly discussed. For instance, paper [1] below could be cited since it provides a unifying and systematic approach summarizing recent results on waveform optimization with also spectral compatibility requirements. Also, it is not easy to understand the novelty of the paper by reading the Introduction. This aspect should be better emphasized by the authors.

[1] A. Aubry, V. Carotenuto, A. De Maio, A. Farina, and L. Pallotta, “Optimization Theory-Based Radar Waveform Design for Spectrally Dense Environments”, IEEE Aerospace and Electronic Systems Magazine, Vol. 31, No. 12, pp. 14-25, 2016.

Response 1: 

Abstract has been modified, Reference to paper [1] has been added

Corresponding modification:

At page 1, line 33.

Point 2:

The utilized notation is always confused. It is difficult to distinguish among vectors and matrices. Sometimes vectors are bold lower case, other times they are lower case with no bold.

Response 2:

I have modified the article in multiple places. Unify the format as much as possible.

Corresponding modification:

In multiple places.

Point 3:

Which is the difference between the two equations in (8)?

Response 3:

Equation (8) is an error generated in editing, and Equation (8) has only one equation.

Corresponding modification:

At page 5, line 136.

Point 4:

Line 165: We assume that only the covariance of the process is known. Please add a footnote recalling that the covariance matrix can be estimated citing also some references that deal with this problem, e.g., [2-5].

[2] I. S. Reed, J. D.Mallett, and L. E. Brennan, “Rapid Convergence Rate in Adaptive Arrays”, IEEE Trans. on Aerospace and Electronic Systems, Vol. 10, No. 4, pp. 853-863, November 1974.

[3] M. Steiner and K. Gerlach, “Fast Converging Adaptive Processors for a Structured Covariance Matrix”, IEEE Trans. on Aerospace and Electronic Systems, Vol. 36, No. 4, pp. 1115-1126, October 2000.

[4] A. Aubry, A. De Maio, and L. Pallotta, “A Geometric Approach to Covariance Matrix Estimation and its Applications to Radar Problems", IEEE Trans. on Signal Processing, Vol. 66, No.4, pp. 907-922, February 2018.

[5] A. De Maio, L. Pallotta, J. Li, and P. Stoica, “Loading Factor Estimation under Affine Constraints on the Covariance Eigenvalues with Application to Radar Target Detection”, IEEE Trans. on Aerospace and Electronic Systems, Vol. 55, No. 3. pp. 1269-1283, June 2019.

Response 4:

We assume that only the covariance of the process is known, which may be unrealistic in practical radar detection environments. Indeed, the corresponding covariance is usually estimated by means of auxiliary samples [22-26].

 Corresponding modification:

At page 7, line 197.

Point 5:

I do not understand why the SINR improvements degrade with increasing K. Please give a motivation to this result.

Response 5:

The CSR involves the () compressed sampling represented by matrix. It is worth observing that if the matrix  is an identity matrix, the radar system illustrated in Fig. 2 becomes conventional radar. If and only if K=N, the corresponding SINR is optimum. But in the CSR case, we have KK, The specific principle is not clear.

Corresponding modification:

Point 6:

I would like to see the comparison of the two proposed methods. Why in the first analyses you consider only the first algorithm, and only then the other? Moreover, since you are in the under-sampled case, i.e. K<N, what happens if you utilized the sample covariance matrix in place of the true one?

Response 6:

In [1], a comprehensive theory of matched illumination waveforms for both deterministic and random targets is presented. Design of matched waveforms based on maximization of both signal-to-noise ratio (SNR) and mutual information (MI) is considered. With this as a motivation, we designed two algorithms for deterministic and random targets respectively. We will use the sample covariance matrix in place of the true one in the next step of the work.

Romero, R.A.; Bae, J.; Goodman, N.A. Theory and Application of SNR and Mutual Information Matched Illumination Waveforms. IEEE Trans. Aerosp. Electron. Syst. 2011, 47, 912–927.

Corresponding modification:

At page 2, line 77.

Reviewer 2 Report

This work studies the SINR maximization in a Compressed sensing radar (CSR), optimizing the transmit waveform, for the purpose of better detection performance without signal recovery. The received signal from the target experiences colored interference as well as noise with known statistics. The target is modeled in two ways. In the former, a deterministic target response is assumed, for which an optimum transmit waveform is derived. As to the latter, the target is modeled with a random function, and an iterative waveform optimization method is given to maximize the SINR. The corresponding comments about the paper are given as follows: 1. As to the references, it is recommended to the authors to enrich the literature review of the paper. For instance it is necessary to include papers: [1] C. Chen and P. P. Vaidyanathan, "MIMO Radar Waveform Optimization With Prior Information of the Extended Target and Clutter," in IEEE Transactions on Signal Processing, vol. 57, no. 9, pp. 3533-3544, Sept. 2009. [2] S. M. Karbasi, A. Aubry, A. De Maio and M. H. Bastani, "Robust Transmit Code and Receive Filter Design for Extended Targets in Clutter," in IEEE Transactions on Signal Processing, vol. 63, no. 8, pp. 1965-1976, April15, 2015. [3] X. Cheng, A. Aubry, D. Ciuonzo, A. De Maio and X. Wang, "Robust Waveform and Filter Bank Design of Polarimetric Radar," in IEEE Transactions on Aerospace and Electronic Systems, vol. 53, no. 1, pp. 370-384, Feb. 2017. 2. It should be better to state that the interference is modeled as signal-independent. 3. The formulation given for the sub-optimal solution to be similar to a conventional radar waveform is not clear. What is the role of the conventional signal in the notations? 4. The presented work is too similar to the work in reference [1] above, without signal-dependent interference and added the matrix Phi. 5. In L191, it is necessary to define a convergence rule. 6. Please give the exact formulation for the LFM signal used in the paper. 7. The parameters used for the sub-optimum method should be given. 8. Why Fig. 4 doesn't contain LFM curve? 9. Fig. 6 and 8 are auto-correlation functions, but they are named time series, why? 10. It is important to explain and justify the fact why we can get better SINR improvements while decreasing \gamma, in deterministic case (Fig. 10)? Also explain the inverse behavior for random case (Fig. 13)? 11. What do the authors mean by the label 'SINR improvement' in Fig.'s 4, 10, and 13? Some other comments through the text body: L83: The utilized sampling waveforms should be carefully addressed and described, or say that it will be discussed later. L86: The use of z in Fig. 1(b) may conflict with the z component of T(z). L98: More clear symbols shall be used to better clarify vectors, matrices and scalar symbols. L98: The parameters should be clearly defined, e.g., What is M and what is N? what is their relation with L? L118: The same sentence is twice referenced to 2 different references, while it seems that 14 is true. L177: equation 28 has typos L211: Identity matrix L267: A subsection title is missing

Author Response

Response to Reviewer 2 Comments

Point 1:

As to the references, it is recommended to the authors to enrich the literature review of the paper. For instance it is necessary to include papers:

[1] C. Chen and P. P. Vaidyanathan, "MIMO Radar Waveform Optimization With Prior Information of the Extended Target and Clutter," in IEEE Transactions on Signal Processing, vol. 57, no. 9, pp. 3533-3544, Sept. 2009.

[2] S. M. Karbasi, A. Aubry, A. De Maio and M. H. Bastani, "Robust Transmit Code and Receive Filter Design for Extended Targets in Clutter," in IEEE Transactions on Signal Processing, vol. 63, no. 8, pp. 1965-1976, April15, 2015.

[3] X. Cheng, A. Aubry, D. Ciuonzo, A. De Maio and X. Wang, "Robust Waveform and Filter Bank Design of Polarimetric Radar," in IEEE Transactions on Aerospace and Electronic Systems, vol. 53, no. 1, pp. 370-384, Feb. 2017.

 Response 1: 

The above references have been added

Corresponding modification:

At page 2, line 65.

Point 2:

It should be better to state that the interference is modeled as signal-independent.

Response 2:

I have modified it as required

Corresponding modification:

At page 3, line 100.

Point 3:

The formulation given for the sub-optimal solution to be similar to a conventional radar waveform is not clear. What is the role of the conventional signal in the notations?

Response 3:

The suboptimal waveform is given by Eq. (22). The corresponding notations have been modified.

 Corresponding modification:

At page7, line 188.

 Point 4:

The presented work is too similar to the work in reference [1] above, without signal-dependent interference and added the matrix Phi.

Response 4:

The idea of this algorithm is similar to that of reference [1] above, but the application fields of this algorithm for compressed sensing radar are different, and the implementation process is also different.

 Corresponding modification:

N/A

At page 3, line 101.

Point 5:

In L191, it is necessary to define a convergence rule.

Response 5:

The transceiver pair (f, h) can be optimized by repeating the steps 1) - 5): Repeat until convergence.

Corresponding modification:

At page 7, line 218-223.

Point 6:

Please give the exact formulation for the LFM signal used in the paper.

Response 6:

It has been modified.

Corresponding modification:

At page 8, line 249.

Point 7:

The parameters used for the sub-optimum method should be given.

Response 7:

Both the optimal and suboptimal solutions are derived in Section 3

Corresponding modification:

Section 3

Point 8:

Why Fig. 4 doesn't contain LFM curve?

Response 8:

Figure 3 has already explained that the suboptimal and optimal waveform has better SINR performances than the LFM waveform in the case of any Gamma (0<Gamma<1). Figure 4 is the SINR improvements provided by optimal and suboptimal waveform are more significant relative to the LFM waveform, not SINR. So Figure 4 doesn't need contain LFM curve

Corresponding modification:

At page 10, line 263.

Point 9:

Fig. 6 and 8 are auto-correlation functions, but they are named time series, why?

Response 9:

Inadvertent errors during translation, It has been modified.

Corresponding modification:

At page 12, line 298. At page 12, line 298.

Point 10:

It is important to explain and justify the fact why we can get better SINR improvements while decreasing \gamma, in deterministic case (Fig. 10)? Also explain the inverse behaviour for random case (Fig. 13)?

Response 10:

The CSR involves the K×N(K≪N) compressed sampling represented by matrix Φ. It is worth observing that if the matrix Φ is an identity matrix, the radar system illustrated in Fig. 2 becomes conventional radar. If and only if K=N, the corresponding SINR is optimum. But in the CSR case, we have K

Corresponding modification:

 At page 6, line 161.

Point 11:

What do the authors mean by the label 'SINR improvement' in Fig.'s 4, 10, and 13?

Some other comments through the text body:

L83: The utilized sampling waveforms should be carefully addressed and described, or say that it will be discussed later.

L86: The use of z in Fig. 1(b) may conflict with the z component of T(z).

L98: More clear symbols shall be used to better clarify vectors, matrices and scalar symbols. L98: The parameters should be clearly defined, e.g., What is M and what is N? what is their relation with L?

L118: The same sentence is twice referenced to 2 different references, while it seems that 14 is true.

L177: equation 28 has typos

L211: Identity matrix

L267: A subsection title is missing

Response 11:

SINR improvement= (SINR_Proposed method)-(SINR_LFM)

L83: It has been modified.

L86: It has been modified.

L98: L211: They have been modified.

L98: The parameters have clearly defined.

L118: It has been modified.

L177: It has been modified.

L211: It has been modified.

L267: It has been modified.

Corresponding modification:

At page 10, line 266.

L83: At page 4, line 115.

L86: At page 3, line 110.

L98: Multiple places

L98: At page 3, line 106-108.

L118: At page 5, line 145.

L177: At page 8, line 210.

L211: At page 6, line 152.

L267: At page 13, line 315.

Reviewer 3 Report

Overall, this paper represents an interesting contribution to waveform-receive filter maximization in compressed domain based on SINR metric. However, before publication, it is my opinion that the following major comments should be addressed by the authors:

1) Abstract – Please rephrase the sentence “There are problems of detection, classification, or estimation and filtering in radar signal processing.” aiming at a more graceful introduction.

2) Abstract – “Two different target models are discussed.” -> “Two different target models are discussed: deterministic and random cases”.

3) Sec. I – “Since the concept of CSR was first proposed in Ref. [5],” -> “Since the concept of CSR was first proposed in the seminal work [5],”

4) Sec. I – “Consequently, the output SINR of CSR can be improved by optimizing it’s transmit waveform. ” -> “Consequently, the output SINR of CSR can be improved by optimizing its transmit waveform.”

5) I would like the authors to discuss, in more detail, how the present work innovates with respect to earlier references [12], [13] and [16].

6) The following related works on waveform-receive filter design (also including polarimetry DoF and security issues) should be discussed for completeness:

[R1] "Robust waveform and filter bank design of polarimetric radar." IEEE Transactions on Aerospace and Electronic Systems 53.1 (2017): 370-384.

[R2] "Full-polarization matched-illumination for target detection and identification." IEEE Transactions on Aerospace and Electronic Systems 38.3 (2002): 824-837.

[R3] "Intrapulse radar-embedded communications via multiobjective optimization." IEEE Transactions on Aerospace and Electronic Systems 51.4 (2015): 2960-2974.

7) Please add a notation paragraph at the end of Sec. I.

8) In this paper, the authors assume known statistics of colored noise, which may be unrealistic in practical radar detection environments. Indeed, the corresponding covariance is usually estimated by means of auxiliary samples, e.g.

[R4] "A CFAR adaptive matched filter detector." IEEE Trans. Aerosp. Electron. Syst 28.1 (1992): 208-216.

[R5] "On the statistical invariance for adaptive radar detection in partially homogeneous disturbance plus structured interference." IEEE Transactions on Signal Processing 65.5 (2016): 1222-1234.

It would be useful if the authors could show an example in which the optimization relies on an estimated covariance.

9) Please discuss the computational complexity of optimization approaches in both deterministic and random cases.

10) Please enrich more future directions paragraph in Sec. 6 (“Conclusions”).

Author Response

Response to Reviewer 3 Comments

Point 1:

Abstract – Please rephrase the sentence “There are problems of detection, classification, or estimation and filtering in radar signal processing.” aiming at a more graceful introduction.

Response 1: 

I have modified as required.

Corresponding modification:

At page 1, line 11-12.

Point 2:

Abstract – “Two different target models are discussed.” -> “Two different target models are discussed: deterministic and random cases”.

Response 2:

I have modified as required.

Corresponding modification:

At page 1, line 20.

Point 3:

Sec. I – “Since the concept of CSR was first proposed in Ref. [5],” -> “Since the concept of CSR was first proposed in the seminal work [5],”

Response 3:

I have modified the corresponding part of the paper.

Corresponding modification:

At page 2, line 42.

Point 4:

Sec. I – “Consequently, the output SINR of CSR can be improved by optimizing it’s transmit waveform. ” -> “Consequently, the output SINR of CSR can be improved by optimizing its transmit waveform.”

Response 4:

I have modified the corresponding part of the paper.

Corresponding modification:

At page 2, line 61.

Point 5:

I would like the authors to discuss, in more detail, how the present work innovates with respect to earlier references [12], [13] and [16].

Response 5:

Optimal linear processing in interference environments in traditional radars uses waveform optimization to maximize output signal-to-noise ratio [15]. Scholars have also proposed distributed compressed sensing methods for complex scenes [12, 16, 17]. In [5], a waveform optimization algorithm based on simulated annealing algorithm is proposed, which can generate waveforms with smaller cross-correlation of targets. Since the recovery algorithm usually includes an iterative process, when the data length is long, the recovery algorithm is also computationally intensive, and this signal processing framework cannot meet the real-time requirements.

Corresponding modification:

At page 2, line 64-69.

Point 6:

The following related works on waveform-receive filter design (also including polarimetry DoF and security issues) should be discussed for completeness:

[R1] "Robust waveform and filter bank design of polarimetric radar." IEEE Transactions on Aerospace and Electronic Systems 53.1 (2017): 370-384.

[R2] "Full-polarization matched-illumination for target detection and identification." IEEE Transactions on Aerospace and Electronic Systems 38.3 (2002): 824-837.

[R3] "Intrapulse radar-embedded communications via multiobjective optimization." IEEE Transactions on Aerospace and Electronic Systems 51.4 (2015): 2960-2974.

Response 6:

It has been modified.

Corresponding modification:

At page 2, line 64-69.

Point 7:

Please add a notation paragraph at the end of Sec. I.

Response 7:

I have modified as required.

Corresponding modification:

At page 3, line 91.

 Point 8:

8) In this paper, the authors assume known statistics of colored noise, which may be unrealistic in practical radar detection environments. Indeed, the corresponding covariance is usually estimated by means of auxiliary samples, e.g.

[R4] "A CFAR adaptive matched filter detector." IEEE Trans. Aerosp. Electron. Syst 28.1 (1992): 208-216.

[R5] "On the statistical invariance for adaptive radar detection in partially homogeneous disturbance plus structured interference." IEEE Transactions on Signal Processing 65.5 (2016): 1222-1234.

It would be useful if the authors could show an example in which the optimization relies on an estimated covariance.

Response 8:

I have modified the conclusions section. Adding some research prospects.

Corresponding modification:

At page 7, line 190.

Point 9:

Please discuss the computational complexity of optimization approaches in both deterministic and random cases.

Response 9:

The computational complexity is not the focus of this paper. Both methods require more than 100 seconds per calculation

Corresponding modification:

N/A

Point 10:

Please enrich more future directions paragraph in Sec. 6 (“Conclusions”).

Response 10:

I have modified the conclusions section. Adding some research prospects.

Corresponding modification:

At page 15, line 360.

Reviewer 4 Report

Some minor comments:

-what is the difference between the two equations in equation 8?

-the logic behind the argument given in equation 14-16 is not super clear (mostly because the notation used in eq.14 and eq.16 appears to contradict itself. I get it but the explanation is not that clear. 

-line 157 “Where” should be “where”. 

- there are two instances of “...where this is the well-known XXX problem” in the manuscript. If a problem and it’s solution are “well-known” then please provide the reader with an appropriate reference where said problem and solution can be found. 

Author Response

Response to Reviewer 4 Comments

Point 1:

what is the difference between the two equations in equation 8?

-the logic behind the argument given in equation 14-16 is not super clear (mostly because the notation used in eq.14 and eq.16 appears to contradict itself. I get it but the explanation is not that clear.

 Response 1: 

Equation (8) is an error generated in editing, and Equation (8) has only one equation.

At equation (14)-(16), the paper appeared in a number of irregularities at the time of writing and has been modified.

Corresponding modification:

At page 5, line 136.

Point 2:

-line 157 “Where” should be “where”.

Response 2:

I have modified the paper in that place.

Corresponding modification:

At page 6, line 159.

Point 3:

- there are two instances of “...where this is the well-known XXX problem” in the manuscript. If a problem and it’s solution are “well-known” then please provide the reader with an appropriate reference where said problem and solution can be found.

Response 3:

I have deleted the word, behind every sentence has been added References.

Corresponding modification:

At page 2, line 58. At page 6, line 170.

Round 2

Reviewer 1 Report

The authors have satisfactorily replied to my previous comments. Now the quality of the paper is improved, therefore it is worth to be accepted for publication. I only suggest to the authors another careful reading of the manuscript to improve further the quality of English.

Author Response

Response to Reviewer 1 Comments

Point 1:

The authors have satisfactorily replied to my previous comments. Now the quality of the paper is improved, therefore it is worth to be accepted for publication. I only suggest to the authors another careful reading of the manuscript to improve further the quality of English.

 Response 1: 

The authors have completely revised the full text and polished the English expression to improve the quality of the paper.

Corresponding modification:

Modified in several places

Reviewer 2 Report

There are still some comments remaining:

One of the recommended references in point 1 is not cited. Some corrections need to be better modified; e.g. point 2. As to point 5, the convergence rule is not clearly formulated. Point 10 is a vital comment and it is necessary to give a logical justification and comprehensive response to it.  

Author Response

Response to Reviewer 2 Comments

Point 1:

One of the recommended references in point 1 is not cited.

Response 1: 

[R2] S. M. Karbasi, A. Aubry, A. De Maio and M. H. Bastani, "Robust Transmit Code and Receive Filter Design for Extended Targets in Clutter," in IEEE Transactions on Signal Processing, vol. 63, no. 8, pp. 1965-1976, April15, 2015.

The above references have been added.

Corresponding modification:

At page 2, line 80.

Point 2:

Some corrections need to be better modified; e.g. point 2.

Response 2:

I have modified and added references.

Corresponding modification:

At page 4, line 138-140.

Point 3:

As to point 5, the convergence rule is not clearly formulated.

Response 3:

The SINR subject to the power constraint via (13), then repeat until convergence. When SINR basically does not change much, (f, h)  can be considered to converge. We can also set the number of iterations according to the accuracy requirements.

 Corresponding modification:

At page 9, line 271-273.

Point 4:

Point 10 is a vital comment and it is necessary to give a logical justification and comprehensive response to it.

It is important to explain and justify the fact why we can get better SINR improvements while decreasing \gamma, in the deterministic case (Fig. 10)? Also explain the inverse behaviour for random case (Fig. 13)?

Response 4:

\gamma is defined as K/N, As \gamma grows, K approaches N, The closer \Phi is full row rank via (13)-(16). If and only if K=N, \gamma=1 the corresponding SINR is optimum, we can get better SINR. Through experiments, we can get better SINR improvements while decreasing \gamma in the deterministic case from Fig. 10. And we can get better SINR improvements while increasing \gamma in the deterministic case from Fig. 13. The specific theoretical reasons still unclear, the next research is needed.

Corresponding modification:

N/A

Reviewer 3 Report

Overall, this paper represents an interesting contribution to waveform-receive filter maximization in compressed domain based on SINR metric. However, before publication, it is my opinion that the following remaining comments should be addressed by the authors, which were almost left unaddressed in the previous review round:

1) The following related works on waveform-receive filter design (also including polarimetry DoF and security issues) should be discussed for completeness:

[R1] "Full-polarization matched-illumination for target detection and identification." IEEE Transactions on Aerospace and Electronic Systems 38.3 (2002): 824-837.

[R2] "Intrapulse radar-embedded communications via multiobjective optimization." IEEE Transactions on Aerospace and Electronic Systems 51.4 (2015): 2960-2974.

2) Please add a mathematical notation paragraph (i.e. summarizing all the mathematical symbols employed) at the end of Sec. I.

3) Please discuss the computational complexity of optimization approaches in both deterministic and random cases. Even a brief discussion would be useful in my opinion.

Author Response

Response to Reviewer 3 Comments

Point 1:

The following related works on waveform-receive filter design (also including polarimetry DoF and security issues) should be discussed for completeness:

 [R1] "Full-polarization matched-illumination for target detection and identification." IEEE Transactions on Aerospace and Electronic Systems 38.3 (2002): 824-837.

 [R2] "Intrapulse radar-embedded communications via multiobjective optimization." IEEE Transactions on Aerospace and Electronic Systems 51.4 (2015): 2960-2974.

Response 1: 

I have modified as required.

Corresponding modification:

At page 2, line 39-43.

Point 2:

Please add a mathematical notation paragraph (i.e. summarizing all the mathematical symbols employed) at the end of Sec. I.

Response 2:

I have modified as required.

Corresponding modification:

At page 3, line 110-119.

Point 3:

Please discuss the computational complexity of optimization approaches in both deterministic and random cases. Even a brief discussion would be useful in my opinion.

Response 3:

I have added the corresponding part of the paper.

Corresponding modification:

At page 16, line 392-394. At page 10, line 312. At page 11, line 321. At page 13, line 356-358.  At page 17, line 378, 383.